# DISTRIBUTIONAL META-GRADIENT REINFORCEMENT LEARNING

**Haiyan Yin, Shuicheng Yan & Zhongwen Xu**[*]
Sea AI Lab

## ABSTRACT

Meta-gradient reinforcement learning (RL) algorithms have substantially boosted the performance of RL agents by learning an adaptive return. All the existing algorithms adhere to the same reward learning principle, where the adaptive return is simply formulated in the form of *expected* cumulative rewards, upon which the policy and critic update rules are specified under well-adopted distance metrics. In this paper, we present a novel algorithm that builds on the success of meta-gradient RL algorithms and effectively improves such algorithms by following a simple recipe, i.e., going beyond the *expected* return to formulate and learn the return in a more expressive form, value distributions. To this end, we first formulate a distributional return that could effectively capture bootstrapping and discounting behaviors over distributions, to form an informative distributional return target in value update. Then we derive an efficient meta update rule to learn the adaptive distributional return with meta-gradients. For empirical evaluation, we first present an illustrative example on a toy two-color grid-world domain, which validates the benefit of learning distributional return over *expectation*; then we conduct extensive comparisons on a large-scale RL benchmark Atari 2600, where we confirm that our proposed method with distributional return works seamlessly well with the actor-critic framework and leads to state-of-the-art median human normalized score among meta-gradient RL literature.

## 1 INTRODUCTION

Meta-gradient reinforcement learning (MGRL) (Xu et al., 2018b; Sutton, 2022) is a family of algorithms that leverage the gradient of the gradient descent update to learn better objectives for reinforcement learning (RL). The MGRL paradigm has achieved substantial performance breakthroughs and dominated the state-of-the-art model-free RL algorithms in various domains, such as Atari (Xu et al., 2018b; Zahavy et al., 2020; Flennerhag et al., 2022), DMLab (Zahavy et al., 2020), and DeepMind Control (Zahavy et al., 2020).

The major algorithmic research on MGRL has been developing towards more and more ambitious targets. For example, it starts from learning two fundamental hyperparameters (i.e., discount factor $\gamma$ and bootstrapping factor $\lambda$) (Xu et al., 2018b), and then goes to self-tuning 20+ hyperparameters in a high-performing RL agent (Zahavy et al., 2020). As a powerful tool for discovering RL semantics from the data, MGRL has been utilized to discover intrinsic rewards (Zheng et al., 2018; 2020), auxiliary tasks (Veeriah et al., 2019), options (Veeriah et al., 2021), etc. Recently, it has even been used to tackle the more flexible form of parameterizing the objective, i.e. using black-box neural networks to learn the objectives from the environment interactions and learning context (Xu et al., 2020; Oh et al., 2020). However, MGRL research has been limited to scalar form, where either meta-parameters are scalar ones (Xu et al., 2018b; Zahavy et al., 2020), or outputs of meta networks are scalars (Xu et al., 2020). This is in line with the traditional definition of returns (Sutton & Barto, 1998), as an expected sum of discounted rewards. However, note that MGRL is not limited to providing RL objectives, while we could also meta-learn a broader concept, e.g., learning rate (Sutton, 1981; Baik et al., 2020; Pinion et al., 2021) and exploration (Gupta et al., 2018; Xu et al., 2018a).

In this work, we investigate an essential step of extending the algorithmic scope of MGRL to distributional methods, where RL algorithms take away the *expectations* in Bellman updates and consider

---

[*]Corresponding author: Zhongwen Xu

a richer formulation of *value distributions* (Bellemare et al., 2017). This is essential for RL updates and even more critical for MGRL which enjoys multiple satisfying properties. Sparse signals are hard to learn in RL, and the bi-level optimization nature of the MGRL makes it extremely difficult to propagate helpful signals to learn meta-parameters. By modeling value distributions, the RL algorithm can naturally produce a rich set of predictions, which provides dense and informative signals for meta-optimization. Moreover, some distributional RL methods come with new optimization properties in the loss form (not available in conventional non-distributional RL methods), e.g., with KL divergence between discrete distributions (Bellemare et al., 2017) or quantile regression (Dabney et al., 2018b). Despite the great accomplishment of distributional RL for non-meta policy learning, the attempt to learn a distributional meta objective has not yet been made by any prior MGRL methods.

This paper aims to develop a novel distributional meta-gradient RL algorithm that can discover meaningful value distributions online. Furthermore, we aim to both methodologically establish a novel distributional meta-gradient framework and find out the intriguing effect of applying distributional meta-gradient techniques on real-world problems, for which we conduct extensive evaluations with a motivating toy domain as well as large-scale end-to-end policy training problems. The evaluation results well demonstrate the effectiveness of our method.

Overall, the contributions of our paper are as follows:

(i) We present a novel meta-gradient variant of the RL method, which considers approximating the distributional return in its value update.

(ii) Our proposed distributional meta-gradient algorithm is general and can be compatible with almost all the existing meta-gradient RL approaches.

(iii) We conduct an extensive empirical study on a toy control domain and the large-scale benchmark Atari 2600. Our results have demonstrated substantial improvements in our method over strong baseline methods on the major evaluation domains.

## 2 RELATED WORK

In recent decades, meta-learning, or *learning-to-learn*, has been explored extensively within the machine learning community (Hospedales et al., 2022; Sutton, 2022), e.g., learning a neural network update rule (Bengio et al., 1990), adapting the learning rate (Sutton, 1992; Schraudolph & Giannakopoulos, 1999), and transferring domain-invariant knowledge (Thrun & Mitchell, 1995). Notably, it has also driven many advances in the RL problems. One direction for such attempts is to learn *meta*-optimizers (Andrychowicz et al., 2016; Wichrowska et al., 2017) or *meta*- policies (Duan et al., 2016; Wang et al., 2017) parameterized by recurrent neural networks, which capture the high-level time-dependent information as meta knowledge. Another direction is gradient-based meta-learning, introduced in Model-Agnostic Meta-Learning (MAML) (Finn et al., 2017a), as well as its followup works (Finn & Levine, 2018; Finn et al., 2017b; Grant et al., 2018; Al-Shedivat et al., 2018). Those works focus on learning a good initialization of the model for one or few-shot multi-task learning with meta-gradients.

In this paper, we consider solving meta-gradient RL problem from a fresh new angle, i.e., optimizing the adaptive *return* as distributions. Policy learning with distributional *return* has been considered in existing RL literature for off-policy learning only. All the existing methods fall into the strand of DQN variants which replace the $n$-step return in DQN with distributions during Q-learning. In (Bellemare et al., 2017), the *return* is formulated as a categorical distribution with fixed values for the base, where the Bellman operator shifts the base values of the distribution, and the target distribution is contracted and projected upon the fixed base. In (Dabney et al., 2018b), the *return* is formulated as quantile distributions, where the algorithm tries to learn the values corresponding to each quantile, alleviating the need for distribution projection or value range approximation. In (Dabney et al., 2018a), the quantiles are approximated by mapping from values drawn from the random distribution (proposals) into quantiles, resulting in an unlimited number of implicit distributions. In (Wang et al., 2019), the proposals in implicit quantiles are replaced by the output of a learned proposal network. In contrast to the aforementioned off-policy distributional RL methods, we tackle a novel problem of learning *adaptive* distributional *return* in on-policy actor-critic algorithms with meta-gradient.

Compared to existing meta-gradient RL methods with scalar *return*, our method applies a distributional return, which is advantageous as (1) it can offer a more informative target to approximate, (2) it naturally provides a richer set of hyperparameters to be tuned with white-box meta-learning, which offers more feedback in the meta-gradient, and (3) approximating such a distributional return naturally results in meaningful auxiliary tasks to facilitate more efficient policy and value training.

## 3 PRELIMINARY

**Quantile Distributional Value Update** In this work, we consider learning the *return* as a quantile distribution. One reason for us to choose quantile distribution rather than any other type of distribution is, the Bellman operator for quantile distribution does not need to perform projection (lose precision) or manually define the *return* boundaries, as C51 (Bellemare et al., 2017) does. Also, our proposed method employs a deterministic prediction function for quantiles which could be conveniently trained, unlike the stochastic mappings employed by IQN (Dabney et al., 2018a), where the stochasticity would bring the noise to the meta gradient and confuse the training of adaptive *return*. We denote the distributional *return* function over each state $s$ as $Z_\theta(s)$, where $\theta$ is its parameter set. With quantile, the distributional *return* can be approximated by a uniform mixture of $N$ Dirac functions, i.e., $Z_\theta(s) := \frac{1}{N} \sum_{q=1}^{N} \delta_{\theta_q}(s)$, where $\theta_q$ is the parameter for the $q$-th Dirac function which can be trained by quantile value update. For a given quantile distribution $Z$ and a quantile $\tau$, the value of the quantile function is denoted as $F_\theta^{-1}(\tau)$. The quantile distribution can be trained by quantile regression loss $\mathcal{L}_\tau^{qr}(\cdot)$ which penalizes overestimation errors with weight $\tau$ and underestimation errors with weight $1 - \tau$, where $\tau \in [0, 1]$:

$$\mathcal{L}_\tau^{qr}(\cdot) = \mathbb{E}_{\hat{Z} \sim Z}\left[\rho_\tau(\hat{Z} - \theta)\right], \qquad \text{where } \rho_\tau(u) = u\left(\tau - \delta_{\{u<0\}}\right), \quad \forall u \in \mathbb{R}. \tag{1}$$

To smooth the quantile loss at zero, we could apply Huber loss over it, i.e.

$$\mathcal{L}_k^{\text{huber}}(u) = \begin{cases} \frac{1}{2}u^2, & \text{if } |u| \leq k; \\ k\left(|u| - \frac{1}{2}k\right), & \text{otherwise.} \end{cases} \tag{2}$$

Thus the Quantile Huber Loss can be derived as follows,

$$\mathcal{J}_{(\tau,k)}^{qr} = |\tau - \delta_{\{u<0\}}| \cdot \mathcal{L}_k^{\text{huber}}(u). \tag{3}$$

**White-box Meta-Gradient RL** White-box meta-gradient RL methods use a bi-level optimization scheme to update important high-level parameters to characterize the *return* and the composed loss function by meta gradient. We denote the high-level parameter set as $\eta$. In the prior work from (Xu et al., 2018b), $\eta = \{\gamma, \lambda\}$; later, loss weights and importance sampling threshold have been added to $\eta$. We denote the RL loss function as $\mathcal{J}(\tau, \theta, \eta)$. During bi-level optimization, the meta-gradient methods first update the model parameter $\theta$ with the gradient $f(\cdot)$ w.r.t $\theta$:

$$f(\tau, \theta, \eta) = \zeta \frac{\partial \mathcal{J}(\tau, \theta, \eta)}{\partial \theta}, \qquad \frac{\partial f(\tau, \theta, \eta)}{\partial \eta} = \zeta \frac{\partial^2 \mathcal{J}(\tau, \theta, \eta)}{\partial \theta \, \partial \eta}, \tag{4}$$

where $\zeta$ is the learning rate and the first-order gradient function $f(\cdot)$ is differentiable w.r.t $\eta$ Then, an alternative batch of transitions $\tau'$ is sampled to optimize the model in the second update stage, where outer objective $\mathcal{J}'$ computed from the updated model parameters $\theta'$ and a reference hyperparameters set $\eta'$ is optimized (e.g., default values for $\eta'$) w.r.t $\eta$:

$$\frac{\partial J'(\tau', \theta', \eta')}{\partial \eta} = \frac{\partial J'(\tau', \theta', \eta')}{\partial \theta'} \cdot \frac{\partial \theta'}{\partial \eta}. \tag{5}$$

Since $\theta'$ is differentiable w.r.t $\eta$, the meta gradient is computed as the gradient of the outer objective $\mathcal{J}'(\tau', \theta', \eta')$ w.r.t $\eta$. Thus meta-gradient RL enables the effective high-level parameters in *learning-to-learn* to compute the adaptive *return* and/or the importance of loss function components.

## 4 DISTRIBUTIONAL META-GRADIENT RL ALGORITHM

**Actor-Critic with Distributional $\lambda$-Return** The principle of RL is to optimize the agent's cumulative rewards, also known as the *return* (denoted as $G$), from the sequence of experiences collected

from the environment, where the experience at time $t$ id denoted as a tuple $\tau_t = (S_t, A_t, R_t, S_{t+1})$ with $S_t$, $A_t$ and $R_t$ denoting the state, action, and reward for the state-action pair at step $t$, respectively. In practice, the *return* can be formulated in various forms. One well-adopted form of *return* is via *bootstrapping*. Let $T$ denote the terminal step and $\gamma$ the discount factor. An example *n*-step *return* (Sutton & Barto, 1998) is defined as:

$$G^{\text{n-step}}(\tau_t) = R_{t+1} + \gamma R_{t+2} + \gamma^2 R_{t+3} + ..., + \gamma^{n-1} v(S_{t+n}), \tag{6}$$

$$G^{\lambda}(\tau_t) = R_{t+1} + \gamma(1 - \lambda)v(S_{t+1}) + \gamma\lambda G^{\lambda}(\tau_{t+1}), \tag{7}$$

where the *return* is the sum of $n$ step rewards and the bootstrapped value for $S_{t+n}$. To better leverage the *return* $G_t$ derived over multiple steps, there is also a $\lambda$-*return*, or TD($\lambda$) *return*, which measures the *return* as the weighted TD error over all time steps (Sutton, 1988), where there is another high-level parameter $\lambda$ introduced to characterize the $\lambda$-*return* function. When $\lambda = 0$, it terminates the bootstrapping from further steps, and when $\lambda = 1$, it promotes pure bootstrapping. White-box meta-gradient RL methods attempt to meta-learn the crucial hyperparameters characterizing the $\lambda$-*return* with a bi-level optimization regime.

We consider the training of actor-critic agents with the critic approximating a quantile distributional *return* predicted by a value function $Z_\theta(S)$, where $S$ is a state, $\theta$ is the parameters of the value function, and $Z_\theta(S)$ follows the *distributional Bellman optimality operator* (Bellemare et al., 2017),

$$\mathcal{T}Z(S) \overset{D}{:=} R(S, A) + \gamma \mathbb{E}_{S' \sim P(\cdot|S,A)} Z(S'), \tag{8}$$

where $U \overset{D}{:=} V$ denotes equality of probability laws, i.e., the random variable U is distributed according to the same law as V. Our proposed distributional adaptive *return* can naturally result in a larger set of self-tuning hyperparameters. The critic is updated towards a canonical *distributional* $\lambda$-return denoted as $\Gamma_t^\lambda$ following a *distributional TD(λ)* algorithm,

$$\Gamma_t^\lambda \overset{D}{:=} R_{t+1} + \gamma_{t+1}(1 - \lambda_{t+1}) Z(S_{t+1}) + \gamma_{t+1}\lambda_{t+1}\Gamma_{t+1}^\lambda,$$

$$\overset{D}{:=} Z(S_t) + \sum_{k=0}^{\infty} \Big( \prod_{j=1}^{k} \gamma_{t+j}\lambda_{t+j} \Big) \underbrace{\Big( R_{t+k+1} + \gamma_{t+k+1}Z(S_{t+k+1}) - Z(S_{t+k}) \Big)}_{\delta_{t+k}}, \tag{9}$$

where $\Gamma_t^\lambda$ is essentially the sum of the predicted *return* distribution $Z(S_t)$ and the backpropagated TD error over all the future steps $\delta_{t+1:\infty}$, where the later could be computed iteratively and weighted by the product of discounting factor and trace in their exponential forms. When training the actor-critic with $\Gamma_t^\lambda$ under a distributed regime like IMPALA (Espeholt et al., 2018) for efficient learning, the training data would become off-policy and we need to perform off-policy correction to get the unbiased gradient. Hence distributional $\lambda$-*return* with off-policy correction in the form of *v-trace* could be defined iteratively as follows,

$$\Gamma_t^\lambda \overset{D}{:=} Z(S_t) + \sum_{k=0}^{\infty} c_{t+k} \Big( \prod_{j=1}^{k} \gamma_{t+j}c_{t+j} \Big) \delta_{t+k}, \quad \text{where } c_m = \lambda_m \Big( \min\big( \bar{c}, \frac{\pi(A_m|S_m)}{\mu(A_m|S_m)} \big) \Big), \tag{10}$$

where $c_m$ is the truncated importance sampling (IS) ratio with $\bar{c}$ being the clipping threshold, and $\pi(\cdot)$, $\mu(\cdot)$ are the present and previous behavior policies, respectively. To update the distributional actor-critic with off-policy correction, we optimize the following objective function:

$$\mathcal{L}_Z = \mathcal{J}_{(\tau,k)}^{qr}(\text{sg}\big(\Gamma^\lambda(S)\big), Z_\theta(S)), \qquad\qquad \mathcal{L}_{\mathcal{H}} = \mathcal{H}\big(\pi_\theta(\cdot|S)\big),$$

$$\mathcal{A}_Z = \frac{1}{N} \sum_{q=1}^{N} \text{sg}\big(\Gamma_{(q)}^\lambda(S) - Z_{\theta_q}(\tau)\big) \cdot \Big( \min\big( \bar{c}, \frac{\pi(A|S)}{\mu(A|S)} \big) \Big), \qquad \mathcal{L}_\pi = -\mathcal{A}_Z(\tau) \cdot \log \pi_\theta(A|S),$$

$$\mathcal{L}_\theta = w_\pi \cdot \mathcal{L}_\pi + w_{\mathcal{H}} \cdot \mathcal{L}_{\mathcal{H}} + w_Z \cdot \mathcal{L}_Z, \tag{11}$$

where $\mathcal{A}_Z$ denotes the advantage computed as the sum over quantile error of the distributional *return*, $\Gamma_{(q)}^\lambda(S)$ denotes the $q$-th quantile in the distributional $\lambda$-*return*, $\mathcal{L}_Z/\mathcal{L}_\pi/\mathcal{L}_{\mathcal{H}}$ and $w_Z/w_\pi/w_{\mathcal{H}}$ are functions and weights for the value/policy/entropy losses respectively, $\text{sg}(\cdot)$ is the *stop-gradient* function and the first term in $\mathcal{L}_\pi$ corresponds to the advantage.

**Distributional Meta-Gradient Update**   Our proposed distributional Meta-Gradient RL algorithm adopts a white-box bi-level optimization scheme to learn adaptive distributional *return* $\Gamma_t^\eta$. We denote the meta parameters characterizing the adaptive *return* $\Gamma_t^\lambda$ and important weights in the actor-critic loss function $\mathcal{L}_\theta$ as a set $\eta$, where $\eta = \{\lambda, \gamma, \alpha, w_\mathcal{H}, w_\pi, w_Z\}$, where $\alpha$ is the weight for the leaky-vtrace in STACX (Zahavy et al., 2020) to control the off-policy correction ratio. Formally, the adaptive return can be formulated as follows,

$$\Gamma_t^\eta \overset{D}{:=} R_{t+1} + \gamma(1-\lambda)\,Z(S_{t+1}) + \gamma\lambda\Big\{\underbrace{Z(S_{t+1}) + \sum_{k=1}^\infty \Big(\prod_{j=1}^k \gamma\lambda\Big)\delta_{t+k}}_{\Gamma_{t+1}^\eta}\Big\},$$

$$\overset{D}{:=} R_{t+1} + \gamma Z(S_{t+1}) + \gamma\lambda \sum_{k=1}^\infty \Big(\prod_{j=1}^k \gamma\lambda\Big)\delta_{t+k}.$$

In other words, we can consider a quantile-specific discounting and bootstrapping operator over the distributions and vectorize the $\lambda$ and $\gamma$ in the set $\eta$. Furthermore, we can also specify a vectorial quantile loss weight $w_Z$. Thus, our method results in a much larger hyperparameter set compared with **STACX**, i.e., $|\eta|$ for **DrMG**, **STACX** and **MG** are $3 \times n_Z + 3$, $7 \times n_{blk}$, and 2, where $n_Z$ and $n_{blk}$ represents the dimension of quantiles and the number of **STACX**'s multi-head blocks.

At the inner loop, the inner loss $\mathcal{L}_\theta^{\text{in}}(\tau, \theta, \eta)$ is first computed to update $\theta$ towards $\theta'$ using the gradient $\partial\mathcal{L}_\theta^{\text{in}}(\tau, \theta, \eta)/\partial\theta$, where $\tau$ is a batch of data sampled to update the inner loss and $\eta$ is drawn from the meta parameters.

$$f(\tau; \theta; \eta) = -\zeta \frac{\partial\mathcal{L}_\theta^{\text{in}}(\tau, \theta, \eta)}{\partial\theta} = -\zeta w_Z \frac{\partial\mathcal{L}_Z}{\partial\theta} - \zeta w_\pi \frac{\partial\mathcal{L}_\pi}{\partial\theta} - \zeta w_\mathcal{H} \frac{\partial\mathcal{L}_\mathcal{H}}{\partial\theta}, \tag{12}$$

where the gradient of the distributional value loss is derived as follows,

$$\frac{\partial\mathcal{L}_Z(\tau, \theta, \eta)}{\partial\theta} = \begin{cases} w_Z \left(\Gamma^\eta(S) - Z_\theta(S)\right)\frac{\partial Z_\theta(S)}{\partial\theta}, & \text{if } |\Gamma^\eta(S) - Z_\theta(S)| \leq k; \\ w_Z \frac{\left(\Gamma_\eta(\tau) - Z_\theta(S)\right)}{|\Gamma_\eta(\tau) - Z_\theta(S)|}\frac{\partial Z_\theta(S)}{\partial\theta}, & \text{otherwise.} \end{cases} \tag{13}$$

At the outer loop, we evaluate the high-level parameters $\eta$ by sampling an alternative batch of transitions $\tau'$ and computing the outer loss $\mathcal{L}_\eta^{\text{out}}(\tau', \theta', \eta')$, where $\eta'$ corresponds to some reference hyperparameters for $\eta$, e.g., the default values predefined like those in other algorithms. The meta gradient to update the meta parameter $\eta$ is defined as follows,

$$f'(\tau', \theta', \eta') = \zeta' \frac{\partial\mathcal{L}_\eta^{\text{out}}(\tau', \theta', \eta')}{\partial\eta} = \zeta' \frac{\partial\mathcal{L}_\eta^{\text{out}}(\tau', \theta', \eta')}{\partial\theta'}\frac{\partial\theta'}{\partial\eta} = f(\tau'; \theta'; \eta')\frac{\partial^2\mathcal{L}_\theta^{\text{in}}(\tau, \theta, \eta)}{\partial\theta\,\partial\eta}, \tag{14}$$

where $\zeta'$ is the learning rate for the outer loss. The meta gradient evaluated at the outer loss interacts with meta parameters through $\frac{\partial\theta'}{\partial\eta}$, which could be further expanded as $\frac{\partial^2\mathcal{L}_\theta^{\text{in}}(\tau, \theta, \eta)}{\partial\theta\,\partial\eta}$ based on the chain rule. With the definition of the distributional return, meta-learning variants become learning the bootstrapping or termination condition over the distributions, rather than scalars, which would bring informative information for return approximation with a greater degree of adaptability. Our proposed Distributional return Meta-Gradient algorithm is denoted as **DrMG**.

## 5   EXPERIMENTS

### 5.1   TWO COLORS GRID-WORLD

**Settings**   We consider a $7 \times 7$ grid-world, where the green-colored agent is assigned to collect the key to open the door at the yellow-colored goal. There are two types of keys; collecting one of them would get a reward of 0.7 and the other 0.3. The agent receives a reward

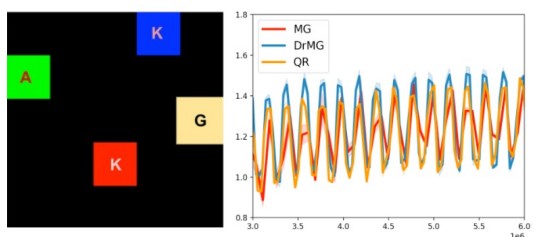

Figure 1: Two colors grid-world environment and the reward curve for **DrMG**, **MG** and a vanilla distributional method **QR** without meta-gradient.

of $+1$ upon successful navigation. After an episode terminates, we randomly generate locations for both keys and the goal. We toggle the reward at every 1e5 steps to investigate the agent's adaptation ability. Thus a key-agnostic optimal agent could collect $+1.5 - r_{pen} * T$ expected return, where $r_{pen}$ and $T$ are the step-wise penalty and episode horizon. We assume that the agent approximating the distributional return should be able to encode the dynamics in its *return* so that it can adapt better compared to the baseline agent approximating the *scalar return*.

**Results** We compare our proposed **DrMG** agent with a meta-gradient baseline algorithm **MG** as well as a vanilla distributional algorithm **QR**, where **QR** is an IMPALA variant which updates a non-adaptive distributional return $\Gamma^\lambda$. The results are shown in Figure 1. We configure **DrMG**, **MG**, and **QR** agents by almost identical hyperparameter sets. Since the RL agents are unaware of the reward toggling during policy training, their adaptation ability is characterized by their efficiency in modeling/approximating the *return* during the training process. We notice that after the reward toggle, **DrMG** achieves peaks with substantially higher *return* compared to that for **MG** and **QR**. Furthermore, **QR** slightly outperforms **MG**, which shows that approximating distributional *return* benefits the task adaptation.

## 5.2 ATARI 2600 200M

**Settings** We testify our method on Atari 2600 Benchmark (Bellemare et al., 2013) under the 200M setting, where the setting aligns with prior works (Xu et al., 2018b; Zahavy et al., 2020). Our agent is developed upon IMPALA (Espeholt et al., 2018), an efficient distributed actor-critic framework. The actor-critic module employs a deep ResNet (He et al., 2016) as the backbone, which is a feed-forward model without LSTM. The network takes 4 grayscale images with size $84 \times 84$ as input. The backbone is connected to a policy-value head where the value head outputs distributional quantiles with a dimension of $N$ and the policy head outputs policy logits with a dimension of $|\mathcal{A}|$. The meta module is a joint set of scalars and vectors, with each vector entry representing a distributional hyperparameter in $\eta$.

For training, we strictly follow the 200M regime without extra data or experience reuse. During training, we sample two batches with identical batch sizes and use them to form two combinations of inner data and outer data to compute the meta-gradient update loop twice. Each task runs with identical hyperparameters and device configurations for hardware and software. We highlight that DrMG brings very little computational overhead; namely, DrMG only brings $|N - 1| * 256$ more parameters compared to MG and it is more computationally efficient than STACX since STACX introduces multiple policy-value heads as its auxiliary task components. All the games employ the same wrapper, for which the action repeat is 4, stick action probability is 0, episodic life is *false*, and the maximum episode length is 108,000. All the tasks adopt the 'no-ops starts' protocol, where at the start of each episode, the agent randomly samples a period to take a dummy action '0' up to 30 steps. We present the environment wrapper setting in Sec. A.1.2 and the detailed model/experiment configurations in Sec. A.2. For comparison, we evaluate the performance on each game in terms of human normalized score (HNS) defined as HNS $= \frac{R_{\text{ours}} - R_{\text{rand}}}{R_{\text{human}} - R_{\text{rand}}}$, where $R_{\text{ours}}/R_{\text{rand}}/R_{\text{human}}$ are the episode rewards for our/random/human expert policies.

**Results** For evaluation over the 57-task benchmark, we compute the *median HNS* for each method. We consider the following five most related baselines: **C51**, **QR-DQN**, Implicit Quantile Network (**IQN**), Self-Tuning Actor-Critic with auXiliary tasks (**STACX**) and **MG**. The first three are off-policy value-based methods which are DQN-variants, while **STACX** and **MG** are white-box meta-gradient RL baselines that are IMPALA-variants, the same as **DrMG**. There is another recently published meta-gradient RL work Bootstrapped Meta-Gradient (**BMG**), which is excluded here for comparison because the work employs a different Atari 2600 benchmark with altered task and model settings, motivated

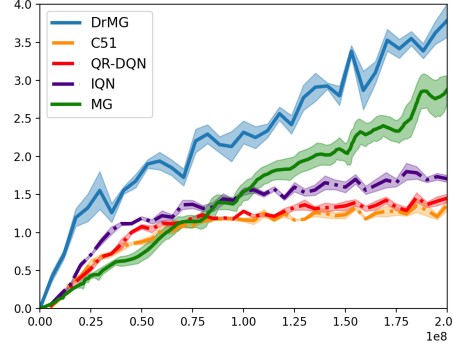

Figure 2: The learning curve in terms of median HNS evaluated on DrMG and its baselines.

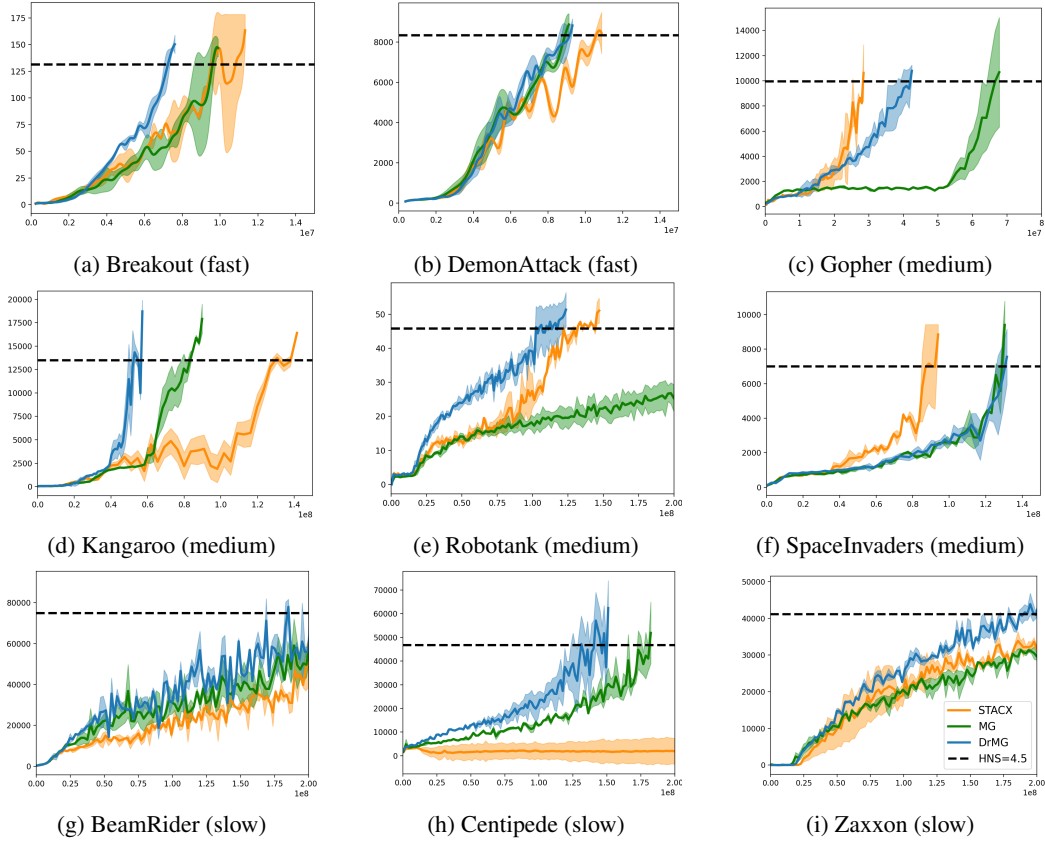

Figure 3: Reward learning curves for our method and meta-gradient RL baselines **MG** and **STACX** evaluated on a representative set of Atari games. Overall, **DrMG** results in stable and efficient progress in reward learning across all the *fast / medium / slow* groups. Its median HNS score on Atari 2600 benchmark is 379%.

to solve orthogonal problems than our work. The detailed explanation is presented in Appendix A.1.2. We show the learning curves for median HNS in Figure 2. We do not present the curve for **STACX** in Figure 2, because the performance of our carefully reimplemented **STACX** drops quite a lot compared to the score in its original paper (even the reimplementation adopts an identical framework, hyperparameters, and model architectures, due to the notorious reproduction issue for Atari 2600 200M series of methods with missing official releases of the baselines experimental code). Nevertheless, the learning curves for **STACX** on representative games can be found in Figure 3.

From the median HNS curves shown in Figure 2, we can first observe that **DrMG** performs substantially better than the conventional distributional RL methods **C51**, **QR-DQN** and **IQN**, which demonstrates updating the distributional *return* adaptively with meta-gradient and the actor-critic framework is more effective than updating the distributional returns by off-policy DQN variants. We also find that **DrMG** brings noticeable performance improvement over the strong white-box meta-gradient RL method **MG**. The median HNS for **DrMG** is 379% and that reported in **MG** is 287%. These results show that optimizing the distributional *return* in our proposed $\lambda$-distributional form is more efficient than learning the scalar *return*.

We present the learning curves of our method, **MG** and **STACX** on a set of representative Atari games in Figure 3. For computational efficiency, the training adopts an *early stopping* policy, where the training terminates after the agent surpasses a fixed HNS standard (450% in our evaluation) for ¿2M steps. We classify the games into three strands {*fast*, *medium*, *slow*}, based on the behavior of the RL agents to hit the designated HNS standard, where *fast*={*Breakout*, *DemonAttack*}, *medium*={*SpaceInvaders*, *Robotank*, *Gopher*, *Kangaroo*} and *slow* = {*BeamRider*, *Zaxxon*, *Centipede*}. Games in *fast*, *medium* and *slow* groups consume $< 10\%$, $30\% - 60\%$ and $\sim 200M$ frames

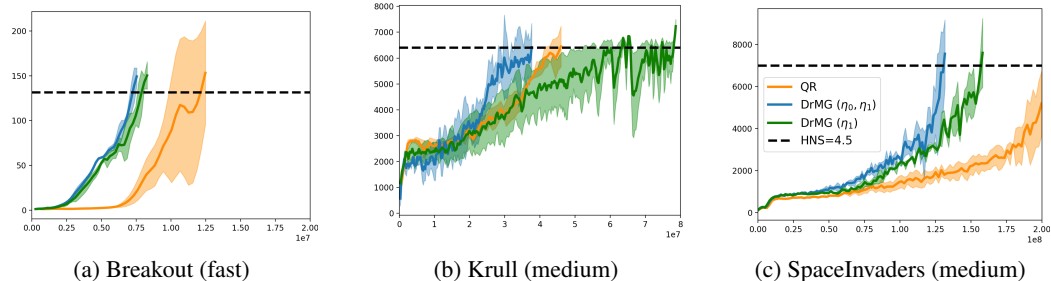

| | |
|---|---|
| (a) Breakout (fast) | (b) Krull (medium) | (c) SpaceInvaders (medium) |

Figure 4: Ablation study results on comparing **DrMG** with a non-adaptive distributional actor-critic baseline **QR**. Baselines learning with non-adaptive distributional *return* performs substantially worse than **DrMG**.

to hit the HNS, respectively. We examine the *fast* group to show that learning the distributional *return* in **DrMG** would not slow down the learning process compared to its counterparts learning scalar *return*. On the classic Atari game *Breakout*, **DrMG** converges noticeably faster than **MG** and **STACX**, and on *DemonAttack*, **DrMG** learns as fast as **MG**. It is also desirable to evaluate learning efficiency for the games from the *medium* group since different methods behave very differently in making progress in those tasks. From the results, we can see **DrMG** progresses significantly faster than the other two methods in *Robotank* and *Kangaroo*. In *Kangaroo*, our method saves almost 1/3 and 1/2 frames compared to **MG** and **STACX**. In *SpaceInvaders*, the progress of **DrMG** is comparable to the others. In the *slow* group, the RL algorithms strive to progress with slow learning paces. All three games are challenging ones for our method as well as the other baselines to achieve high HNS. So, their performance on them is crucial to derive the median HNS score, i.e., pulling back anyone could potentially lower the median HNS. We can observe that **DrMG** leads to fairly stable performance among those in *slow* group. In *BeamRider*, *Centipede* and *Zaxxon*, **DrMG** achieve better asymptotic rewards than the baselines. It is also shown in Figure 3 that the performance of **STACX** is not ideal, e.g., irregular performance in *DemonAttack* and *Kangaroo*.

## 5.3 ABLATION STUDY

We perform ablation studies to investigate the effects of the following designs in our method: (i) the *adaptive* distributional *return* vs. *non-adaptive* distributional *return*; (ii) $\lambda$-distributional return vs. $n$-step distributional return; (iii) vector hyperparameter $\eta$ vs. scalar $\eta$; (iv) adopting BMG's bootstrapped inner steps on top of our method, where the results of ablation study (iv) are presented in appendix A.3 for detailed comparison/discussion.

***Adaptive* vs. *non-adaptive* distributional *return*** We investigate the effects of adaptive/non-adaptive distributional return by comparing **DrMG** with two ablated baselines. The meta parameter set is split into two groups $\eta = \{\eta_0, \eta_1\}$, where $\eta_0 = \{\lambda, \gamma, \alpha\}$ and $\eta_1 = \{w_{\mathcal{H}}, w_{\pi}, w_Z\}$. The first baseline only meta-update $\eta_1$, denoted as **DrMG($\eta_1$)** (complete version is denoted as

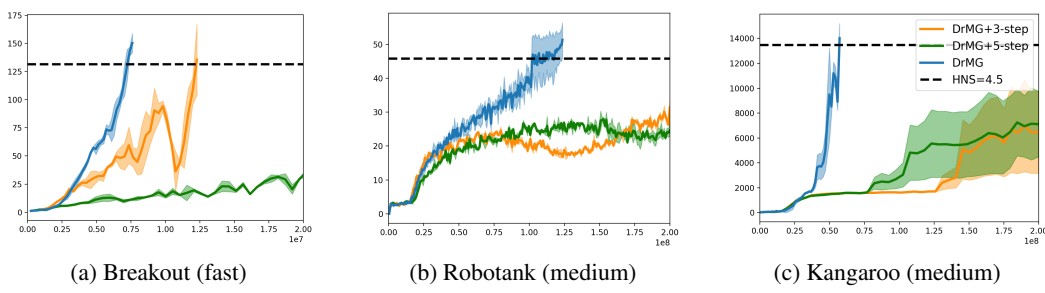

| | |
|---|---|
| (a) Breakout (fast) | (b) Robotank (medium) | (c) Kangaroo (medium) |

Figure 5: Ablation study results on comparing the $\lambda$-return used in **DrMG** with an ablated baseline **DrMG+n-step** which replaces the $\lambda$-distributional return with an $n$-step distributional return, where we consider two cases $n = 3$ and $n = 5$.

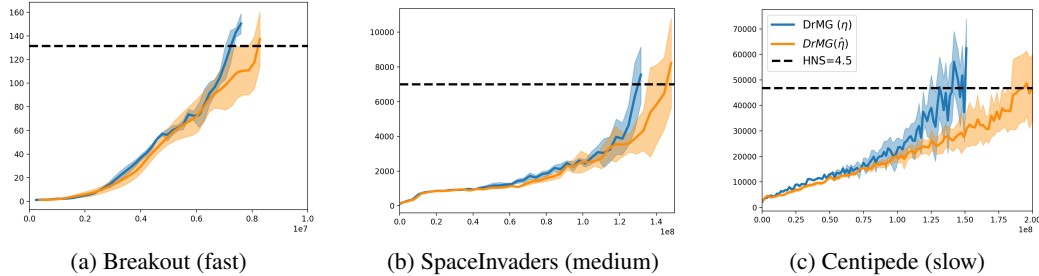

(a) Breakout (fast)   (b) SpaceInvaders (medium)   (c) Centipede (slow)

Figure 6: Ablation study results for comparing our proposed vector hyperparameter set $\eta$ (**DrMG($\eta$)**) with a scalar set hyperparameters $\bar{\eta}$ (**DrMG($\bar{\eta}$)**). **DrMG($\hat{\eta}$)** performs comparable to **DrMG($\eta$)** the *fast* and *medium* games and underperforms **DrMG($\eta$)** in the *slow* game.

**DrMG($\eta_0, \eta_1$)**). The second baseline is **QR**, a non-meta method equivalent to **DrMG({})**. We compare them in *Breakout*, *Krull*, and *SpaceInvaders*. The results are shown in Figure 4. We first notice that both versions of *non-adaptive* distributional $\lambda$-*return* work considerably well with the actor-critic IMPALA framework (outperform their corresponding DQN variants by large margins). Using **QR** can learn as smoothly as **DrMG** in *Krull* but for the other two domains, it underperforms **DrMG**, which shows the effectiveness of using meta-gradient to optimize distributional *return*. **DrMG($\eta_1$)** outperforms **QR** in 2/3 games, but it underperforms **DrMG($\eta_0, \eta_1$)**, which demonstrates the superiority of meta-learning adaptive *return* in distributional meta-gradient algorithms.

**$\lambda$-distributional *return* vs. $n$-step distributional *return*** We investigate the effects of $\lambda$-/$n$-step distributional return by comparing **DrMG** with a meta-gradient RL baseline that replaces the $\lambda$-distributional *return* by an $n$-step *return* computed over the *return* distributions, denoted as **DrMG+n-step**. We consider two values for $n$ in evaluation, where $n = 3$ and $n = 5$, i.e., resulting in baselines **DrMG+3-step** and **DrMG+5-step**. We adopt three domains, *DemonAttack*, *SpaceInvaders*, and *Centipede*. The results are shown in Figure 5. We can observe that replacing $\lambda$-return with $n$-step return substantially pulls down the rewards even in the *fast* game *Breakout*. This further confirms that backpropagating the distributional TD errors from $\lambda$-return is highly efficient to formulate the *adaptive* distributional *return* target.

**Vector hyperparameters $\eta$ vs. scalar hyperparameters $\bar{\eta}$** We then investigate the effects of vector hyperparameters $\eta$ vs. scalar hyperparameters $\bar{\eta}$ by training **DrMG** with two kinds of adaptive distributional *return* parameterizations. When using scalar hyperparameters $\bar{\eta}$, the *return* for each quantile defined in $\Gamma^\lambda$ shares the same discounting factor $\gamma$, trace $\lambda$, as well as cost for the quantile regression loss. When using a vector form of $\eta$, each quantile head enjoys a quantile-specific parameterization for the adaptive *return*. We evaluate **DrMG($\eta$)** and **DrMG($\bar{\eta}$)** in *Breakout*, *SpaceInvaders* and *Centipede*, where the baselines are most distinguishable with **DrMG($\bar{\eta}$)**. The results are presented in Figure 6. We observe that **DrMG($\hat{\eta}$)** performs fairly well in *Breakout* as well as *SpaceInvaders*, where other ablated baselines easily fall in the latter case. However, for the *slow* game *Centipede*, **DrMG($\bar{\eta}$)** noticeably underperforms its vector version **DrMG($\bar{\eta}$)**.

## 6 CONCLUSION

We present a meta-gradient algorithm with distributional return named **DrMG**, which employs a critic that learns adaptive distributional return with parameters to be trained by meta-gradient derived from a white-box bi-level optimization scheme. We demonstrate that the quantile distributional *return* can seamlessly work with distributed actor-critic algorithms and learning such a distributional adaptive distributional *return* can offer more meaningful signals to update the meta parameters. We show **DrMG** is able to outperform its most related non-meta-learning baseline IMPALA and several decent RL baselines on a motivating Two-Colors Grid-World domain and the challenging video game benchmark Atari 2600. Our method is also very general and the distributional *return* we introduce hereby can be adapted to work with alternative meta-gradient RL methods, with the quantile distributional *return* being able to be flexibly replaced with other forms of distributions. We leave the work of integrating more variants of actor-critic algorithms or distributional *returns* in **DrMG** as our future work. It is also worth considering applying **DrMG** on more practical RL domains, such as pure value-based RL, offline R, and continuous control domains.

## ETHICS AND REPRODUCIBILITY STATEMENT

Our work tackles synthetic digital game-playing applications and therefore we believe our work is free from ethical problems. We present details on the distributional framework, Atari wrapper settings, and model hyperparameters in Appendix. We also include the code for the two important algorithmic components, networks, and the two-level meta-update step function in our work.

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
