# OpenReview forum: "Distributional Meta-Gradient Reinforcement Learning"
_ICLR.cc/2023/Conference — ICLR 2023 poster_

### Official Review · Reviewer_LNTi · 2022-10-23

**Confidence:** 4
**Correctness:** 4
**Technical Novelty And Significance:** 2
**Empirical Novelty And Significance:** 2
**Recommendation:** 6

**Clarity, Quality, Novelty And Reproducibility:**

# Detailed Comments (Clarity, Quality, Novetly, Reproducibility)

-   Section 1, Paragraph 1 (Meta-gradient RL): It is limiting to say
    that meta-gradient RL only provides better objectives. While this is
    true in Xu et al. 2022, meta-gradients can also be used to
    meta-learn hyperparameters unrelated to the target or objective,
    such as the learning rate or epsilon for exploration.

-   Section 1, Paragraph 2: "This is essential for RL updates and even
    more critical for meta-gradients that enjoy multiple satisfying
    properties"

    It is not clear what "this" refers to, but if you mean value
    distributions then this is not true nor demonstrated in any
    meta-gradient RL work.

-   "On the other hand, distributional RL methods have good optimization
    properties in the loss form"

    I do not think this is true either. While there are worse options
    than the KL divergence for stability, that does not mean that the KL
    loss in distributional RL has particularly good proprerties in
    comparison to non-distributional RL.

-   Section 2, Paragraph 3: This paragraph is key and really helped me
    understand the goal of this paper more than the abstract or
    Section 1. I would encourage the authors to rework this into the
    introduction.

-   Section 3 " the Bellman operator for quantile distribution does not
    need to perform projection (lose precision) or manually define the
    return boundaries, like C51 (Bellemare et al., 2017) does." This
    seems at odds with the claim that distributional RL is a good method
    for meta-learning because of the additional hyperparameters it
    requires.

-   Section 4 ( Actor-Critic with Distributional λ-Return): this
    majority of this subsection seems better suited to the preliminaries
    section. It is not clear which parts are new contributions or
    relating to previously established work on meta-gradient
    actor-critic.

-   Section 4: "Our proposed distributional adaptive return can
    naturally boost up the scope of self-tuning hyperparameters in the
    meta-gradient RL algorithm."

    Not clear what this means exactly, but the principle of self-tuning
    hyperparameters is not limited in scope in any fundamental way. Using meta-gradients to tune specific hyperparameters related to distributional return is merely an application.

-   Section 5.1 (Two Colors Grid): While the results are compelling, in
    the sense that DrMG adapts quicker than MG, it is not clear whether
    this is due to Distributional meta-gradients or just the
    distributional base algorithm. Unfortunately, the appendix is short
    on details here.

-   Section 5.2 (Atari 2600 200M): Again, the results here are
    compelling, albeit difficult to ascertain their true significance.
    Atari 200M is a large and difficult task, and the reported
    performance seems promising. But, as an evaluation benchmark, it is
    quite limited due to its size and inability to run many seeds with
    confidence intervals.

-   Section 5.3 (Ablating adaptive vs non-distributional return): It is
    true that an important ablation study would compare the importance
    of adaptive and non-adaptive distributional return. However, an even
    more important baseline would be a meta-gradient algorithm that uses a distributional return but does
    not take a meta-gradient through the distributional hyperparameters. Without this very important baseline, it's difficult to tell how important meta-gradients are for distributional RL in particular.

# Minor Comments

-   Section 3 "computed from the updated model parameters θ"

    Should this be: "computed from the updated model parameters θ'"?

    If not, you should be clear what θ′ is, for example
    θ′ = θ − f(τ, θ, η).

-   Section 4, Distributional Meta-Gradient (usage of α): earlier you
    used α as a step-size so it is confusing to use it for the leak
    v-trace as well.

-   Section 4, "for our method; with STACX and MG it would be 3 × N + 3,
    7 × 3 and 2, respectively." What is N here?

-   Section 5: "We toggle the reward at each 1e5 steps to investigate
    the agent’s adaptation ability. " What does this mean exactly? If
    the blue key gives a reward of 0.7 and the red key gives a reward of
    0.3, does toggling switch the color-reward correspondence.

-   Section 5.1 title: "GIRD-WORLD" -> "GRID-WORLD"



**Strength And Weaknesses:**

# Strengths

-   Strong empirical results, versus the baselines presented. This is,
    however, somewhat overshadowed by STACX missing as a baseline. It is
    also to be expected that adapting more hyperparameters allows for a
    more flexible learner as it is more general than the baselines
    compared against.

-   The contribution is clear: both meta-gradient RL methods and
    distributional RL are strong methods in current deep RL. Combining
    them together is a clear and interesting research direction.

# Weaknesses

-   Some important baselines and experimental details are missing that
    would help contextualize the merit of distributional meta-gradients.
-   Overall lacking in novelty. While the contribution is clear and
    interesting, combining meta-gradients and distributional RL is
    rather obvious as both methods are orthogonal.
-   The experiments are somewhat limited. Atari 200M is a large and
    difficult problem, and success on this benchmark is commedable.
    Being a large benchmark, however, limits the accuracy of the
    conclusions because of the high degree of variability. Although the
    results are averaged over 5 runs, no error bars are reported and the
    ablation on various games is somewhat ad hoc.


**Summary Of The Paper:**

This paper proposes to combine white-box meta-learning methods and
distributional RL. The main benefit of this combination is that the
distributional return can not be adaptive, and perhaps allow for more
effective RL agents. In particular, the authors propose a meta-gradient
through the parameters that define the quantile and its loss. The
distributional meta-update is said to be compatible will almost all
meta-gradient RL work. Experiments are first conducted on a small
non-stationary grid world problem, but the majority of the results are
on Atari 200M, where they beat distributional baselines and a
representative meta-gradient baseline in aggregated score. They note,
however, that the SOTA meta-gradient baseline - STACX - is difficult to
reproduce and do not compete directly.



**Summary Of The Review:**


# Decision

While there is obvious appeal in combining two currently separate but
successful RL methods (that of distributional and meta-gradient RL), I
think the paper as a whole does not provide enough insight into
challenges and benefits of such a combination. The paper clearly
demonstrates that the proposed method (DrMG) is able to outperform
distributional baselines and a meta-gradient baseline. The paper also
demonstrates that adaptive distributional returns are important in some
Atari games. As a whole though, the Atari benchmark makes sound
empirical comparison difficult due to the fact that no statistical
evidence is presented. In addition, some important baselines are missing
(STACX, which cannot be reproduced to the reported performance as well
as a distributional meta-gradient baseline that does not adapt the
distributional hyperparameters). As the paper currently stands, I am
leaning towards weak reject with the possibility of increasing to weak
accept if some of the gaps in the baselines are addressed.

---

> ### Author Response · Authors · 2022-11-17
> **Author Response to Reviewer LNTi (part 2/n)**
>
> > **Q7**: The majority of the subsec. (Actor-Critic with Distributional $\lambda$-Return) seems better suited to the preliminaries sec. It is not clear which parts are new contributions or related to previously established work on meta-gradient actor-critic.
>
> **A**: The distributional TD-$\lambda$ return we derived in Eq (9) is a new yet important property in our method: (1) **conventional distributional RL methods do not consider learning $\lambda$-return** (mostly fit to optimizing $n$-step return as DQN variants); (2) **our work is the first one to formulate $\lambda$-return with value distributions in the actor-critic algorithm**, where the subsect. is a new yet essential part of the solution; (3)  **we also empirically demonstrate our proposed distributional  $\lambda$-return is more effective** than conventional distributional non-$\lambda$ and $n$-step return (as shown in Figure 5).
>
> ***
>
> > **Q8**: Sec 4: "Our proposed distributional adaptive return can naturally boost up the scope of self-tuning hyperparameters in the meta-gradient RL algorithm." Not clear what this means exactly, but the principle of self-tuning hyperparameters is not limited in scope in any fundamental way. Using meta-gradients to tune specific hyperparameters related to distributional return is merely an application.
>
> **A**: The reviewer is right. We have revised the statement as
>
> > Our proposed distributional adaptive return can naturally result in a larger set of self-tuning hyperparameters.
>
> ***
> > **Q9**: While Two Colors Grid results are compelling, in the sense that DrMG adapts quicker than MG, it is not clear whether this is due to Distributional meta-gradients or just the distributional base algorithm.
>
> **A**: We thank the reviewer for the valid concern. **We have added a baseline that only uses a distributional return without meta-gradient (denoted as QR)**). Please find the learning curve for it updated in **Figure 1**. The result shows that the performance is promising when only using the distributional return since QR slightly outperforms MG.  However, the performance of QR is inferior to DrMG, which shows the effectiveness of learning *adaptive* distributional return.
>
> ***
> > **Q10**: Atari benchmark is quite limited due to its size and inability to run many seeds with confidence intervals.
>
> **A**: We update **all the results with confidence intervals over 3 random seeds**, which is the same as the number of random seeds adopted by STACX. Overall, DrMG demonstrates considerable stability and leads to state-of-the-art median HNS among meta-gradient RL algorithms.
>
> ***
> > **Q11**: An even more important baseline would be a meta-gradient algorithm that uses a distributional return but does not take a meta-gradient through the distributional hyperparameters.
>
> **A**: We have conducted an comparison on the suggested baseline and  **the result is presented in Figure 4**. Specifically, we can split the hyperparameter set into two groups $\eta=$ \{ $\eta_0, \eta_1$ \}, where the group $\eta_0$ corresponds to parameters characterizing adaptive return  $\eta_0$ = \{ $\lambda, \gamma,  \alpha$ \}, while $\eta_1$ corresponds all the other parameters outside of $\eta_0$, i.e., $\eta_1 =$ \{ $w_{\mathcal{H}}, w_\pi, w_Z$ \}.
>
> The suggested meta-gradient with non-adaptive return baseline is denoted as **DrMG**($\eta_1$), while our original method can be considered as **DrMG**($\eta_0,\eta_1$).  We can find that training meta-gradient RL agent with the non-adaptive return is substantially inferior to training with the adaptive return, confirming it is essential to (1) **meta-learn** (2) **adaptive return**.
>
> ***
> > **Q12**: Sec 4, Distributional Meta-Gradient (usage of α):
>
> **A**: Updated $\alpha$ to $\zeta$.

---

> > ### Author Response · Authors · 2022-11-17
> > **Author Response to Reviewer LNTi (part 3/n)**
> >
> > > **Q13**: Sec 4, "for our method; with STACX and MG it would be 3 × N + 3, 7 × 3 and 2, respectively."  What is N here?
> >
> > **A**: The numbers have been revised as $3\times n_{Z} + 3$, $7\times n_{blk}$ and 2, where $n_{Z}$ and $n_{blk}$ represents the dimension of quantiles for DrMG and number of multi-head blocks for STACX, respectively.
> >
> > ***
> > > **Q14**: "We toggle the reward at each 1e5 steps..."  Does toggling switch the color-reward correspondence?
> >
> > **A**: Yes.
> >
> > ***
> > > **Q15**: The Atari benchmark makes sound empirical comparison difficult due to the fact that no statistical evidence is presented. In addition, some important baselines are missing (STACX, which cannot be reproduced to the reported performance as well as a distributional meta-gradient baseline that does not adapt the distributional hyperparams).
> >
> > **A**: The conclusion STACX cannot be reproduced can be partially revealed in the learning curves for our implementation of STACX shown in Figure 3, i.e., there are noticeable biases between reproduced and reported performance of STACX which makes the median HNS hard to reach, though we carefully tuned the algorithm under its suggested hyperparameters. We will release the code for baselines MG and STACX together with DrMG after the paper is accepted.

---

> ### Author Response · Authors · 2022-11-17
> **Author Response to Reviewer LNTi (part 1/n)**
>
> We thank Reviewer LNTi for providing insightful comments. We have updated the learning curves with error bars. We have also added a new QR baseline in the toy gridworld domain (Figure 1) as well as a non-adaptive meta-gradient baseline DrMG($\eta_1$) (Figure 4). We have also conducted careful proofreading to correct the grammar/definition errors in our paper.
> ***
> > **Q1**: lacking in novelty, combining meta-gradients and distributional RL is rather obvious as both methods are orthogonal.
>
> **A**: **We highlight that the effort of combining two RL directions is non-trivial**. In our work, we expand the algorithmic scope of meta-gradient RL to consider a new form of adaptive return. This kind of idea has never been explored by either the meta-gradient RL community or the distributional RL community. Furthermore, we present solid yet general methodological formulations to solve the problem. We believe our work could potentially become an important stepstone to inspire and cultivate many future works to work in this direction.
> ***
> > **Q2**: high degree of variability..., no error bars are reported and the ablation on various games is somewhat ad hoc.
>
> **A**: We have added the error bars to **all the curves in our paper** (refer to Figure 1/2/3/4/5/6/7). Our method demonstrates **clear performance margins** over the baselines with **considerable stability** compared with the baselines.
> ***
> > **Q3**: It is limiting to say that meta-gradient RL only provides better objectives. Meta-gradients can also be used to meta-learn hyperparameters unrelated to the target or objective, such as the learning rate or epsilon for exploration.
>
> **A**: Thanks for the thoughtful comments. We have added claims in the Introduction section as
>
> > However, note that MGRL is not limited to providing RL objectives, while we could also meta-learn a broader concept, e.g., learning rate [1][2][3] and exploration [4][5].
>
> - [1] Adaptation of learning rate parameters (Technical Report’81).
>
> - [2] Meta-Learning with Adaptive Hyperparameters (NeurIPS’20).
>
> - [3] ETA Prediction with Graph Neural Networks in Google Maps (Arxiv’21).
>
> - [4] Learning to Explore via Meta-Policy Gradient (ICML’18).
>
> - [5] Meta-Reinforcement Learning of Structured Exploration Strategies (NeurIPS’18).
> ***
> > **Q4**: Sec 1, Paragraph 2: "This is essential for RL updates and even more critical for meta-gradients that enjoy multiple satisfying properties". It is not clear what "this" refers to, but if you mean value distributions then this is not true nor demonstrated in any meta-gradient RL work.
>
> **A**: The “this” refers to “the effort of extending the algorithmic scope of MGRL to distributional methods". If it is still unclear, we could discuss it further.
>
> ***
> > **Q5**: I don’t think “distributional RL methods have good optimization properties in the loss form" is true. There are worse options than the KL divergence for stability, that does not mean that the KL loss in distributional RL has particularly good properties in comparison to non-distributional RL.
>
> **A**: The reviewer is right as there do exist extreme cases in distributional RL that could be hard to optimize, and the claim should have referred to more specific cases. We have carefully revised the statements into
> > Moreover, some distributional RL methods come with new optimization properties in the loss form (not available in conventional non-distributional RL methods), e.g., with KL divergence between discrete distributions [6] or quantile regression [7].
>
> - [6] A Distributional Perspective on Reinforcement Learning (ICML’17).
>
> - [7] Distributional Reinforcement Learning with Quantile Regression (AAAI’18).
>
> ***
> > **Q6**: Sec 3 " the Bellman operator for quantile distribution does not need to perform projection (lose precision) or manually define the return boundaries, like C51 (Bellemare et al., 2017) does." This seems at odds with the claim that distributional RL is a good method for meta-learning because of the additional hyperparameters it requires.
>
> **A**: It is true that not all distributional RL methods suit well to work with meta-gradient, e.g., C51. But we think it is not necessary to require each distributional RL method to work well with the meta-gradient to prove the idea of distributional meta-gradient is valid. The fact a subset of distributional RL algorithms does not fit the meta-gradient should not diminish the merit of distributional meta-gradient, since there are methods like Quantile Regression that fits fit well to our framework, and we have empirically demonstrated the strong performance of such a combination. Therefore, we should not let the few counter-examples deny the promise of distributional meta-gradient RL.

---

> ### Comment · Area_Chair_BXJM · 2022-11-21
> **Any comments to the responses from the authors?**
>
> Dear Reviewer LNTi,
>
> Thank you very much for your detailed review.  The authors have provided responses to your concerns together with additional experimental results.  How did they change your evaluation, particularly on the comparison against baselines?

---

> > ### Comment · Reviewer_LNTi · 2022-11-22
> > **Primary Addressable Concern Has Been Addressed**
> >
> > Dear Area Chair,
> >
> > While the paper is overall incremental, I think the idea is well-executed. Especially with the additional baseline, which shows that learning to adapt the distribution is important in certain Atari games. I will increase my score as a 6, but I find the paper still lacking in the following regards:
> >
> > - Primary evaluation on Atari lacks statistical power. While their results indicate significance in certain games, this is only over 3 runs and so does not really present a strong statistical conclusion.
> >
> > - Incremental: using meta-gradients to adapt distributional hyperparameters is relatively straight forward. However, I do not consider this a strong con.
> >
> > As such, I cannot increase my score further than a 6. But the contribution seems solid.

---

> > > ### Author Response · Authors · 2022-11-23
> > > **Response to Additional Comments from Reviewer LNTi**
> > >
> > > Dear Reviewer LNTi,
> > >
> > > We sincerely thank you for providing valuable comments on our response. We want to elaborate more on the 3 random seeds issue which you are mostly concerned with.
> > >
> > > At the current stage, we adopted 3 random seeds for the experiment because **all** the **related meta-gradient RL** works referred to in our paper adopted **<=3 random seeds** for the shared Atari evaluation benchmark  (e.g., MG used 1 seed, and STACX/BMG used 3 seeds). Among the **related distributional RL** works, C51/QR-DQN/FQF used 3 seeds, and *only* IQN used 5 seeds. On the other hand, there are well-known deep RL methods like DQN and Rainbow DQN that run with only 1 seed, and the distributed method IMPALA that our work was based on claimed it used 3 seeds for ablation comparisons. **To conclude**, we are following *exactly the same practice* of other RL literature in our evaluation of the Atari 57 games' full benchmark.
> > >
> > > We also wish to highlight that with 3 seeds, we actually evaluate our method on $57 \times 3$ games, i.e., **171 runs of the Atari experiments** (under 200M) for the single benchmark. So the number of seeds we employ should be considered differently from that for other experiments with a single or a small subset of tasks since *every median score* data point in our case is a statistic from 171 runs of experiments. Also, as we have noted earlier, deriving such a median score would *consume quite a number of computational resources* (~4.22 GPU days for running 3 seeds), which is considered to be expensive even for large research organizations with tremendous computation resources like DeepMind.
> > >
> > > To resolve the reviewer’s concern about the strength of the statistical conclusions in our paper, **we could update all the results with 10 random seeds** in the final version of our paper if the reviewer thinks it would be helpful. It will approximately take 10+ more GPU days to finish. If you have any further questions, please let us know.

---

### Official Review · Reviewer_K6pA · 2022-10-24

**Confidence:** 4
**Correctness:** 4
**Technical Novelty And Significance:** 3
**Empirical Novelty And Significance:** 3
**Recommendation:** 8

**Clarity, Quality, Novelty And Reproducibility:**

+ Clear description of the justification of the proposed approach and how it fits in the state of the art.
+ The primary novelty of the paper is to approximate the distributional value in the value update of the RL, and finding an efficient method to do this.

**Strength And Weaknesses:**

+ Convincing justification that the estimation through a quantile distribution represented through a uniform mix of Dirac's deltas.
+ In the evaluation on both a toy problem and on Atari games, the overall performance shows a clear improvement over the comparable algorithms.

- When replacing a numerical value with a distribution, the reader would be naturally interested in how this distribution looks like. Is it Gaussian, skewed, fat tailed, multimodal distribution? The paper makes no effort to investigate this.

**Summary Of The Paper:**

The paper is considering meta-gradient RL, algorithms that consider the gradient of the gradient descent update to learn better objectives. The authors are proposing an algorithm called DrMG which is based on two foundational choices: (a) the use of a quantile distribution for the return and (b) the use of a white-box meta-gradient RL model where the loss function depends on two sets of parameters, the low-level parameters theta and the high-level parameters eta. The algorithm used for RL is a actor-critic class algorithm, with the critic approximating a quantile distribution.

**Summary Of The Review:**

A novel contribution to the field of model free RL providing faster learning and higher performance than comparable algorithms on the standard benchmarks.

---

> ### Author Response · Authors · 2022-11-17
> **Author Response to Reviewer K6PA**
>
> We thank Reviewer K6pA for providing encouraging comments. We wish to succinctly highlight the major changes we have made to improve our paper:
> - We have added error bars to **all the learning curves**.
> - We have added a more ablation baseline **QR** to the toy gridworld domain, to compare the effectiveness of updating distributional return with/without *meta-gradient* (Figure 1).
> - We have also added two ablative baselines (1. meta-gradient with non-adaptive return; 2. DrMG with BMG’s bootstrapping at its inner loop), and results are available in Figures 4 & 7 from our paper.
>
> ***
> >**Q1**: Important baselines and experimental details are missing
>
> **A**: The learning curves for STACX are available on all the highlighted games in Figure 3. The result reveals that there are noticeable gaps between reproduced and reported performance of STACX in quite a few games. Such gaps would make the median HNS hard to reach. We have carefully tuned STACX based on its released hyperparameters. We will release the code for baselines MG and STACX together with DrMG upon acceptance.
>
> ***
> >**Q2**: When replacing a numerical value with a distribution, the reader would be naturally interested in how this distribution looks like. Is it Gaussian, skewed, fat tailed, multimodal distribution?
>
> **A**: The reviewer is right that the form of distributions is important to understand our work. The existing work **C51** [1] adopts a categorical distribution with learnable x-axis values, **QR-DQN**  [2]  and **IQN**  [3]  learns a quantile distribution, while there are also other methods like **MMDDQN**  [4]  that learns an unstructured distribution represented by static samples. The discussion on distribution forms is available in the second paragraph of the Related Work section.
>
> - [1] A Distributional Perspective on Reinforcement Learning (ICML’17).
>
> - [2] Distributional Reinforcement Learning with Quantile Regression (AAAI’18).
>
> - [3] Implicit Quantile Networks for Distributional Reinforcement Learning (ICML’18).
>
> - [4] Distributional Reinforcement Learning via Moment Matching (AAAI’21).

---

### Official Review · Reviewer_ApMf · 2022-11-03

**Confidence:** 4
**Correctness:** 3
**Technical Novelty And Significance:** 3
**Empirical Novelty And Significance:** 3
**Recommendation:** 6

**Clarity, Quality, Novelty And Reproducibility:**

- Clarity: The paper is overall well-written and easy to follow in most places.
- Quality: The experimental results are very promising, and the ablation study is quite helpful in explaining the effect of the adaptive return and the lambda return.
- Novelty: The idea of applying a meta-gradient method to distributional RL has never been substantiated or evaluated before. That said, the overall algorithm design appears to be a rather direct combination of meta gradient and quantile-based distributional RL.
- Reproducibility: This shall be acceptable given that the critical parts of the pseudo code and the hyperparameters are provided in the appendix.



**Strength And Weaknesses:**

**Strength**

- This paper provides the first distributional attempt of meta-gradient RL.
- The empirical results are quite promising in Atari 2600 benchmarks.


**Weakness**

- The proposed algorithm appears to be a direct combination of distributional actor-critic and the standard meta-gradient method. The novelty of the algorithm is somewhat limited.
- Comparison with baselines: This paper empirically compares DrMG mainly with the standard MG and a more recent method STACX. However, the most recent baseline Bootstrapped Meta Gradient (BMG) was excluded for comparison. The paper mentioned that “... which is excluded here for comparison because it alters some important experimental settings for 200M evaluation to boost the performance.” Despite this, it still appears feasible to adjust the setting of BMG to ensure a fair comparison. I do not fully get the difficulty here. More justification would be needed.


Additional issues: Some explanations of the methodology are incorrect or confusing. For example:
- The distributional Bellman operator in Eq (8) appears incorrect (cf. Eq. (5) in [Bellemare et al., 2017]).
- There shall be some typos in Eq. (14), e.g., the learning rates are missing.
- The last quality of Eq. (14) would require explanation in more detail.


**Summary Of The Paper:**

In the existing meta-gradient studies, the adaptive return is simply formulated in the form of expected cumulative rewards. In this paper, the authors address the meta learning problem of the distributional actor-critic method with quantile distributions by considering the distributional $\lambda$ return and apply the standard meta-gradient method to self-tune a set of hyperparameters. Some of the hyperparameters are quantile-specific, which offers a larger scope of hyperparameters compared to the prior works. This new algorithm outperforms the existing meta-gradient methods in Grid-world and Atari games, achieving the state-of-the-art performance.

**Summary Of The Review:**

This paper proposes an effective meta gradient method to boost the performance of quantile-based distributional RL. Despite that the idea is not very novel, the experimental results on Atari are very promising, and the ablation study verifies the efficacy of the adaptive return.

---

> ### Author Response · Authors · 2022-11-17
> **Author Response to Reviewer ApMF**
>
> We thank Reviewer ApMF for providing insightful comments. We have added additional results comparing DrMG with a bootstrapped version of DrMG (named DrMG+Bootp) in Appendix 3.
>
> ***
> >**Q1**: The proposed algorithm appears to be a direct combination of distributional actor-critic and the standard meta-gradient method. The novelty of the algorithm is somewhat limited.
>
> **A**: We wish to highlight that our proposed method is not a direct combination of the two methods and it comes with considerable technical novelty. We highlight that our work expands the algorithmic scope of meta-gradient RL and the effort of combining two directions is non-trivial. This kind of idea has never been explored by either the meta-gradient RL community or the distributional RL community. Furthermore, we present solid yet general methodological formulations to facilitate the problem solution. We believe our work could potentially become an important stepstone to inspire and cultivate many future works in this direction.
>
> ***
> >**Q2**: The most recent baseline Bootstrapped Meta Gradient (BMG) was excluded for comparison… it still appears feasible to adjust the setting of BMG to ensure a fair comparison. I do not fully get the difficulty here. More justification would be needed
>
> **A**: **[Discussion - empirical]** First, we wish to clarify that the effort of adjusting the setting of BMG to add it as a baseline for a fair comparison with our work is non-trivial, mainly due to the following factors:
> - BMG uses a different ALE wrapper, which makes the task different from the one adopted by our work. The main change is changing the grayscale input (84×84)  into RGB input (160×210×3).
> - BMG uses a much larger ResNet, where the channel depth changed from (16, 32, 32) to (64,128,128,64).
> - BMG uses replay, wherein in each batch, the replay to online ratio is 2:1, whereas MG/STACX/DRMG do not employ replay. BMG with replay is also trained under a smaller batch size than MG/STACX/DRMG, where the batch sizes for the former and latter algorithms are 18 and 32, respectively.
>
> The last point is somewhat more troublesome than the first two because it requires **heavy engineering effort** to fully integrate a reply buffer into the code base of MG/STACX/DrMG when BMG does not have open-source code. Meanwhile, running the BMG experiment with experience replay and a small batch size would significantly increase the run time for the method. It also remains unclear whether an IMPALA variant **with replay** could be considered a fair baseline to compare with the other meta-gradient methods in our work (MG/STACX/DrMG) that **do not perform data reuse**.
>
> **[Discussion - methodological]** Note that from the methodological point of view, BMG and our work tackle orthogonal directions in meta-gradient RL problem, i.e., BMG is innovated on the **solver level**, focusing on how to better optimize the bi-level optimization with bootstrapping, whereas our work focuses on modeling the adaptive distributional return which is solved by a standard meta-gradient solver (with two-step inner-outer update).
>
> **[Additional results for the new BMG baseline]** However, we implemented BMG's core techniques of bootstrapping upon our method, creating a new variation of DrMG which is referred to as the **Bootstrapped DrMG (denoted as DrMG+Bootp)**. We compare DrMG+Bootp with DrMG on a group of six games and show the results **in Figure 7**.  Overall, the performance of DrMG+Bootp is on par with DrMG on two fast learning games  (*Breakout*, *DemonAttack*) and a medium learning game (*Krull*), but it underperforms DrMG on the other three games which consist of a medium learning game *Gopher* as well as two slow learning games *Centipede* and *BeamRider*. We also present a detailed discussion on DrMG+Bootp in Appendix 3.
>
> We have also added the aforementioned insights on BMG in Appendix 1.1.2 of our paper.
>
> ***
> >**Q3**: The distributional Bellman operator in Eq (8) appears incorrect (cf. Eq. (5) in [Bellemare et al., 2017]).
>
> **A**: We have updated the conventional equation to the distributional equation.
>
> ***
> >**Q4**: There shall be some typos in Eq. (14), e.g., the learning rates are missing.
>
> **A**: We have checked Eq. (14) and corrected the mistake (missing $\partial$ symbol for numerators).  We clarify that there is no learning rate issue because, for the rightmost part of the formula, the learning rate is absorbed by the function $f(\tau’,\theta’;\eta')$. We also added explanations on the derivation of this equation after its appearance.

---

> ### Comment · Area_Chair_BXJM · 2022-11-21
> **Any comments to the responses from authors?**
>
> Dear Reviewer ApMf,
>
> Thank you very much for your informative review.  The authors have provided clarifications and responses to your concerns.  How did they change your evaluation, particularly on the novelty and the justification for the exclusion of BMG?

---

### Official Review · Reviewer_s6vn · 2022-11-03

**Confidence:** 4
**Correctness:** 3
**Technical Novelty And Significance:** 3
**Empirical Novelty And Significance:** 3
**Recommendation:** 6

**Clarity, Quality, Novelty And Reproducibility:**

- Clarity

  - The proposed method is clearly described and easy to follow.

  - The experiments section can be improved with more details and accurate descriptions.

- Quality

  - The quality of this paper is ok but can be polished a little bit more.

- Novelty

  - To the best of my knowledge, this paper is the first to combine distribution RL with meta-gradient RL.

- Reproducibility

  - A lot of implementation details are presented in the appendix, but it would be great if the authors could release the code to the public.

**Strength And Weaknesses:**

The strength and weaknesses of this paper are both significant.

## Strength

- The idea of combining distributional RL and meta-gradient RL is natural and intuitive.

- The proposed algorithm DrMG seems to achieve great performance on the Atari benchmark compared to existing meta-gradient RL methods.

- The ablation study considers several important design choices.



## Weakness

The weaknesses mainly come from the experiments.

- The toy experiment "Two Colors Grid-World" is very confusing to me. Specifically,

  - In Sec 5.1, this paper mentions "We toggle the reward at each 1e5 steps to investigate the agent’s adaptation ability." What is the exact meaning of "toggle the reward at each 1e5 steps"? According to the appendix, it seems the key collection reward values toggle between the two keys. However, if this is the case, then why a key-agnostic optimal agent could collect a +1.35 expected return? I think it should be 1.5.

  - In figure 1, the dashed line denoting the key-agnostic agent is neither 1.5 nor 1.35. What is the exact number of it and how do you get it?

  - Why the reward is modified in the middle of training? This is an uncommon rare practice in RL. The authors claim that "Since the RL agents are unaware of the reward toggling during policy training, their adaptation ability is characterized by their efficiency in modeling the return during the training process." I do not really understand this point. Why the "efficiency in modeling the return" needs to be shown in such a weird setting?

- The experiments on Atari are not conducted properly

  - First, it seems all the curves come without std. It is unconvincing if the experiments are only run with one random seed, especially when the curves look very unstable. I understand the computation cost of those experiments is high, but it is still necessary to try a least a few more random seeds. In STACX, they use 3 random seeds.

  - It is unclear why the experiments shown in figure 3 adopt an early stopping strategy but the experiments shown in figure 2 do not. Given that the full curves (training with 200M) should be available according to figure2, why not show the full curves in figure 3?

- Minor points

  - The legends in the figures should be consistent. For example, "MG" in figure 2 but "MetaGrad" in figure 3.

**Summary Of The Paper:**

This paper proposes a Distributional return Meta-Gradient algorithm (DrMG) algorithm, which combines distributional RL (like C51) and meta-gradient RL. Specifically, the authors formulate a distributional return in a meta-gradient RL setup and derive a meta-update rule to learn the adaptive distributional return with meta-gradients. Empirically, the proposed algorithm is compared with the existing meta-gradient RL methods on 57 Atari tasks.

**Summary Of The Review:**

Overall, I feel this paper proposes a good approach that combines distributional RL and meta-gradient RL. However, the empirical evaluation of their approach is not done properly. Given that the idea itself is not extremely fancy, I feel proper evaluation is very important in this case. Therefore, I cannot recommend acceptance.


=========
Update after the rebuttal period:

I previously leaned toward rejection. My main concern is that the experiments are not sufficient to support their claims fully. Given that this paper's technical novelty is not super significant (combining distribution RL and meta-gradient RL), I feel proper evaluation is very important in this case.

However, during the rebuttal period, the authors provided additional experiment results, including more random seeds for each experiment, more baselines, and more ablations. The additional results make the empirical evaluation much stronger. Therefore, my main concern is addressed, and I lean toward acceptance.

---

> ### Author Response · Authors · 2022-11-17
> **Author Response to Reviewer s6vn (1/n)**
>
> We thank Reviewer s6vn for providing insightful comments. We would like to clarify our toy experiment domain and have added error bars to the learning curves. We have also carefully polished the writing of our paper.
>
> ***
> >**Q1**: What is the exact meaning of "toggle the reward at each 1e5 step"? According to the appendix, it seems the key collection reward values toggle between the two keys. If this is the case, then why a key-agnostic optimal agent could collect a +1.35 return? it should be 1.5.
>
> **A**:  We apologize for this typo. The actual expected return for key-agnostic agent should be $1+(0.7+0.3)/2 - r_\text{pen}*T$, where $r_\text{pen}$  is the *step-wise reward penalty* and $T$ is the *expected episode length*.
>
> ***
> >**Q2**: In figure 1, the dashed line denoting key-agnostic agent is neither 1.5 nor 1.35. What is exact number and how you get it?
>
> **A**: We apologize for the confusing value for the horizontal line. The value was computed as subtracting step-wise penalty from a wrong reference 1.35. We have removed the horizontal dashed line to address the issue.
>
> ***
> >**Q3**: Why the reward is modified in middle of training? This is an uncommon rare practice in RL. Why "efficiency in modeling return" needs to be shown in such a weird setting?
>
> **A**:  The example is not an uncommon case. The toy example is motivated by the tabular grid-world domain employed by a related work Bootstrapped Meta-Learning. Our other highly related meta-gradient works, such as MG [1] and FRODO [2], also employ such toy domains to examine the adaptive property of the return.
>
> Since the compared methods are only different in their return forms and/or update rule, the evaluation thus testifies *the efficiency of the compared algorithms in approximating the return*.
>
> Furthermore, introducing toy domain(s) for proof-of-concept evaluation could be helpful to better *examine the validity of the motivation* and *the effectiveness of the proposed method in policy training*. Note that we also  **added a QR baseline in the toy domain** to compare the effect of approximating the quantile return with/without meta-gradient (following the Reviewer LNTi’s comment **Q9**).
>
> - [1] Meta-gradient reinforcement learning (NeurIPS’18).
>
> - [2] Meta-Gradient Reinforcement Learning with an Objective Discovered Online (NeurIPS’20).
>
> ***
> >**Q4**:  All (Atari) curves come without std. It is unconvincing if experiments are only run with one seed, esp. when the curves look very unstable. STACX uses 3 seeds.
>
> **A**:  We thank the reviewer for the thoughtful concern. We have updated **all the curves in our paper** with standard deviation bars, where we use 3 random seeds. It shows that our method **remains to outperform the baselines by clear margins** with the uncertainty statistics.
>
> ***
> >**Q5**:  It is unclear why experiments shown in figure 3 adopt an early stopping strategy but experiments shown in figure 2 do not. Given that full curves (training with 200M) should be available according to figure2, why not show the full curves in figure 3?
>
> **A**:  We wish to clarify that it is possible to derive the median HNS curve in Figure 2 with early stopping. We define the early stopping strategy as keeping HNS>450% (a value far above our estimated median HNS so that we can have much fewer games run for 200M fully) for > 2M steps. For example, there are four games (B, C, D, E) where B and E stop earlier than the max steps budget 200M, (quit at time T_B and T_E) but C and D run till the end of 200M. After a game early stops,  we assume its reward remains unchanged thereafter (e.g., after 100M /75M steps, the reward for game B/E remains >450%). Thus, as long as there exist at least one top 29 games (i.e., the median over 57 games) that quit not before 200M, the median HNS curve would be the full curve.
>
> B:|--------->T_B=100M
>
> C:|------------------->|T_C=200M
>
> D:|-------------------->|T_D=200M
>
> E:|------> T_E=75M
>
> We also wish to highlight that the early stopping strategy is a very efficient one to consider for Atari 2600 experiments with two major merits: (1) early stopping helps to *save machine hours* significantly, e.g., given our average frames-per-second (FPS) is ~ 20K, running top 29 Atari 2600 games with 3 seeds with/without early stopping would consume GPU hours of **~ 33.75 * 3 (~ 4.22 days)**/**~ 80.56 * 3 (~ 10 days)**, reducing **> 50%** of computational resources; (2) the learning curves with early stopping could help distinguish better the performance of agents on fast/medium learning games, e.g., Breakout and DemonAttack.
>
> However, we will try to provide full training curves upon acceptance to eliminate the confusion.
>
> ***
> >**Q6**: The legends should be consistent: "MG" in figure 2 but "MetaGrad" in figure 3.
>
> **A**: Corrected. We have also aligned *IMPALA+QR* with *QR*.

---

> > ### Author Response · Authors · 2022-11-17
> > **Author Response to Reviewer s6vn (2/n)**
> >
> >
> > >**Q7**: It would be great if the authors could release the code to public.
> >
> > **A**: In Appendix, we have added detailed code for several important components, i.e., base, meta networks (A.4.3), and learner’s two-level optimization for Atari 2600 (A.4.2), which are crucial for understanding DrMG. Complete runnable code with config file will be released upon acceptance.

---

> > > ### Comment · Reviewer_s6vn · 2022-11-24
> > > **Reply to the authors**
> > >
> > > Thanks for the author's response. I appreciate the additional experiment results, which make the claims more convincing. I have raised my rating. See my additional comments below.
> > >
> > > ### Regarding the toy example
> > >
> > > I am still not convinced that this toy example is not an uncommon case. The fact that the example is used by other papers does not directly show this example is a common case. Can you give a real-world example that the reward being changed in the middle of training? Also, I checked the meta-gradient paper [1] but I did not find this toy example. Can you explicitly point out where did they use this example?
> > >
> > > ### Regarding the early stopping
> > >
> > > I buy your explanation. It would be great if you added those details to the appendix or somewhere.

---

> > > > ### Author Response · Authors · 2022-11-29
> > > > **Response to Additional Comments**
> > > >
> > > > Dear Reviewer s6vn,
> > > >
> > > > Thank you very much for providing thoughtful feedback on our author response.
> > > >
> > > > First, we would like to clarify that many real-world reinforcement learning problems come with non-stationary rewards (refer to [a][b][c]). Therefore, the study of reinforcement learning problems under non-stationary rewards is a meaningful and crucial task to consider for the reinforcement learning community.
> > > >
> > > > [a] A Survey of Reinforcement Learning Algorithms for Dynamically Varying Environments (ACM Computing Surveys’2021).
> > > >
> > > > [b] Factored Adaptation for Non-Stationary Reinforcement Learning (Neurips‘2022).
> > > >
> > > > [c] Optimizing for the Future in Non-Stationary MDPs (ICML’2020).
> > > >
> > > > ***
> > > > Please find our response to your detailed questions as follows.
> > > >
> > > > >**Q1** The fact that the example is used by other papers does not directly show this example is a common case. *Can you give a real-world example that the reward being changed in the middle of training?*
> > > >
> > > > **A**:  To answer this question, we first quote two environments adapted from such real-world scenarios mentioned in [c] as follows:
> > > >
> > > > **(1/2) Non-stationary Diabetes Treatment**:
> > > >
> > > > This environment is based on an open-source implementation (Xie, 2019) of the FDA approved Type-1 Diabetes Mellitus simulator (T1DMS) (Man et al., 2014) for treatment of Type-1 Diabetes. Each episode consists of a day in an in-silico patient’s life. Consumption of a meal increases the blood-glucose level in the body and if the blood-glucose level becomes too high, then the patient suffers from hyperglycemia and if the level becomes too low, then the patient suffers from hypoglycemia. The goal is to control the bloodglucose level by regulating the insulin dosage to minimize the risk associated with both hyper and hypoglycemia.
> > > >
> > > > However, the insulin sensitivity of a patient’s internal body organs vary over time, inducing non-stationarity that should be accounted for. In the T1DMS simulator, we induce this non-stationarity by oscillating the body parameters (e.g., insulin sensitivity, rate of glucose absorption, etc.) between two known configurations available in the simulator.
> > > >
> > > > [Xie, 2019]  URL https://github.com/jxx123/simglucose.
> > > >
> > > > [Man et al., 2014] The UVA/PADOVA type 1 diabetes simulator: New features.
> > > >
> > > > **(2/2) Non-stationary Recommender System**:
> > > >
> > > > In this environment, a recommender engine interacts with a user whose interests in different items fluctuate over time. In particular, the rewards associated with each item vary in seasonal cycles. The goal is to maximize revenue by recommending an item that the user is most interested in at any time.
> > > >
> > > >
> > > >
> > > > Additionally, we wish to highlight examples of other important real-life applications with non-stationary rewards as follows:
> > > >
> > > > - Reinforcement learning for **stock trading** where the buying and selling prices for the stocks can vary from time to time during each trading day.
> > > >
> > > > - Decision-making systems that **predict production throughputs for oil manufacturing** companies where both the cost for the raw manufacturing materials and the sales price for the products can be highly non-stationary.
> > > >
> > > > - In a **multi-agent reinforcement learning environment**, when an agent executes the same action under the same state, the reward could be non-stationary as the reward function is depending on the actions of other agents (e.g., different opponents appearing at alternative locations in the map).
> > > >
> > > >
> > > > We hope the aforementioned examples could help justify the importance of studying problems like two-colors gridworld which come with non-stationary rewards. Please let us know if there is any further concern on evaluating algorithms with non-stationary domains.
> > > >
> > > > ***
> > > > >**Q2** Also, I checked the meta-gradient paper [1] but I did not find this toy example. Can you explicitly point out where did they use this example?
> > > >
> > > > **A**: We wish to clarify that we do not mean [1] also uses the identical toy domain as we do.
> > > > Our statement is as follows,
> > > > >  Our other highly related meta-gradient works, such as MG [1] and FRODO [2], also employ such toy domains to examine the adaptive property of the return.
> > > >
> > > > In [1], two toy example prediction tasks are introduced to show meta-gradient can self-tune $\gamma$ and $\lambda$, respectively, towards some interpretable directions to better solve the tasks,  e.g., resulting in high discount value for states with positive rewards and low discount value for those with negative rewards (top and bottom figures in Figure 1(a) from [1]).
> > > >
> > > > We wish to highlight that all the follow-up works for [1] no longer adopt the same toy domains because we do not need to prove the same concept again. In our work, we consider adopting the two-colors gridworld domain to justify the effectiveness of approximating *adaptive* distributional return, for which tasks with non-stationary reward property are more straightforward to consider than stationary domains, to let the algorithms adapt against.

---

> > > > > ### Comment · Reviewer_s6vn · 2022-12-06
> > > > > **Response**
> > > > >
> > > > > Thank the authors for the detailed explanation. The examples given by the authors look convincing to me. I do not have further questions.

---

> ### Comment · Area_Chair_BXJM · 2022-11-21
> **Any comments to the responses from authors?**
>
> Dear Reviewer s6vn,
>
> Thank you very much for your informative review.  The authors have provided responses to your concerns.  How did they change your evaluation, particularly on your concerns about experiments?

---

> > ### Author Response · Authors · 2022-11-23
> > **Follow-up with Reviewer s6vn**
> >
> > Dear Reviewer s6vn,
> >
> > We sincerely thank you for providing insightful comments to help improve our paper.
> >
> > We have summarized major updates to our paper as follows:
> > - [Atari & Gridworld curves]: all curves have been updated with standard deviation computed from 3 random seeds (we will update the results with **10 random seeds** in the final version of our paper).
> > - [Gridworld baseline]: we have added an additional baseline **QR** that learns *distributional return* without meta-gradient update (available in Figure 1).
> > - [Atari baselines]: we have added **two ablative baselines**: (a) meta-gradient with non-adaptive return; (b) DrMG with BMG’s bootstrapping at its inner loop, where the results are available in Figures 4 & 7.
> > - [Release code]: we have provided **additional implementation details** of DrMG: (a) detailed code for the base and meta networks (available in Section A.4.3); implementation of the inner and outer update steps from the learner agent (available in Section A.4.2). Complete code with config files for DrMG/MG/STACX will be released after our paper is accepted.
> >
> > We appreciate it if you could let us know if there is any remaining concern that we could discuss further on.

---

### Decision · Program_Chairs · 2023-01-20

**Decision:**

Accept: poster

**Justification For Why Not Higher Score:**

The proposed approach is a combination of distributional RL and meta-gradient RL, and its algorithmic novelty is rather limited and may not appeal to a broad audience.

**Justification For Why Not Lower Score:**

The reviewers were unanimous in their opinion that the proposed approach is novel, provides new insights, and its effectiveness is strongly supported by experiments.

**Metareview: Summary, Strengths And Weaknesses:**

This paper proposes a general framework of leveraging distributional reinforcement learning (RL) in meta-gradient RL.  The adaptive distributional return learned by the distributional RL method can offer information that is useful in updating meta-parameters b meta-gradient RL.  While the combination of distributional RL and meta-gradient RL might not be super innovative (which is the main weakness of the paper), the proposed method has been validated extensively in the experiments with Atari, which also gives empirical insights into this combination.  This innovation in the fundamental idea together with strong empirical support constitute the major strength of the paper.



**Note From Pc:**

if the above contains the word "oral" or "spotlight" please see: "oral" presentation means -> notable-top-5% and "spotlight" means -> notable-top-25%. As stated in our emails, we are disassociating presentation type from AC recommendations